# The incidence and survival after in-hospital cardiopulmonary cerebral resuscitation in end-stage kidney disease patients: A nationwide population-based study

**Chia-Hung Yang[1], Jia-Jin Chen[2,3], Jih-Kai Yeh[4], George Kuo[2], Cheng-Chia Lee[2], I-Chang Hsieh[4], Ming-Jer Hsieh[4], Ya-Chung Tian[2], Chih-Hsiang Chang[2,3]***

**1** Department of Cardiology, New Taipei Municipal TuCheng Hospital, Chang Gung Memorial Hospital and Chang Gung University, **2** Kidney Research Center, Department of Nephrology, Chang Gung Memorial Hospital, Linkou Medical Center, Taoyuan, Taiwan, **3** School of Medicine, College of Medicine, Chang Gung University, Taoyuan, Taiwan, **4** Department of Cardiology, Chang Gung Memorial Hospital, Linkou Medical Center, Taoyuan, Taiwan

* franwisandsun@gmail.com

**Data Availability Statement:** All relevant data are within the manuscript and its Supporting Information files.

## Abstract

### Background

This study analyzed the survival and protective predictors of in-hospital cardiopulmonary cerebral resuscitation (CPCR) to potentially help physicians create effective treatment plans for End-stage kidney disease (ESKD) patients.

### Methods

We extracted the data of 7,116 ESKD patients who received their first in-hospital CPCR after initial dialysis between 2004 and 2012 from the National Health Insurance Research Database. The primary outcome was the survival rate during the first in-hospital CPCR. The secondary outcome was the median post-discharge survival.

### Results

From 2004 through 2012, the incidence of in-hospital CPCR decreases from 3.97 to 3.67 events per 1,000 admission days (*P* for linear trend <0.001). The survival rate for the first in-hospital CPCR did not change significantly across the 9 years (*P* for trend = 0.244), whereas the median survival of post-discharge survival increased significantly from 3.0 months in 2004 to 6.8 months in 2011 (*P* for linear trend <0.001). In addition, multivariable analysis identified older age as a risk factor and prior intracardiac defibrillator (ICD) or cardiac resynchronization therapy defibrillator (CRT-D) implantation as a protective factor for in-hospital death during the first in-hospital CPCR.

### Conclusion

The incidence of in-hospital CPCR and the duration post-discharge among ESKD patients improved despite there being no significant difference in the survival rate of ESKD patients

**Funding:** C-HC was supported by the Ministry of Science and Technology (106-2314-B-182A-118-MY3) and grants from Chang Gung Memorial Hospital, Taiwan (CMRPG3F1653, CMRPG5H0162). C-HY was supported by Chang Gung Memorial Hospital, Taiwan (CORPG3G0761).

**Competing interests:** The authors have declared that no competing interests exist.

after CPCP. Either ICD or CRT-D implantation may be advisable for ESKD patients with a high risk of sudden cardiac death.

## Background

The incidence of in-hospital cardiac arrest in the general population has grown in the past decade, in line with aging societies and increasingly complex comorbidities [1]. The survival-to-discharge rate of cardiopulmonary cerebral resuscitation (CPCR), the possibility of favorable neurologic recovery, and functional status warrant the most consideration for patients following cardiac arrest. Although the advanced cardiac life support guidelines already define a cardiac arrest algorithm for use in resuscitation, the effects of survival on in-hospital cardiac arrest are rarely reported [2].

Despite the decreased overall death rate in dialysis patients since 2001, as found by a United States Renal Data System report, sudden cardiac arrest and arrhythmia remain the leading cause of death in end-stage kidney disease (ESKD) patients [3–5]. One of the noteworthy features of ESKD patients with cardiac arrest is the catastrophic outcome, with 30-day and 1-year survival rates of 32% and 15%, respectively [6], higher than those of the general population [7]. Most related research has focused on risk factors for sudden cardiac death or cardiac arrest in ESKD patients [8–13]. Some studies have suggested that beta blockers may have a preventive or prognostic effect for cardiac arrest in ESKD patients and that the effects of intracardiac defibrillators (ICDs) are inconsistent [14–17]. According to Saeed et al. [7], patient age of less than 65 years, receiving cardiac catheterization, and hospital teaching status are predictor factors for discharge to home. Beta blockers and ACEI/ARB but not age are associated with higher survival in ESKD patients with cardiac arrest in a cohort study by Pun et al. [15]. Studies examining the prognostic factor for ESKD patients with cardiac arrest after CPCR appear to present controversial findings and are small in number; in addition, no large population-based study on the topic has been conducted in an Asian country.

Therefore, our aims in performing this large-scale population-based cohort study were to 1) determine the incidence of in-hospital CPCR among ESKD patients, 2) investigate the trends in survival after in-hospital CPCR among ESKD patients, and 3) verify the protective factors related to successful CPCR among ESKD patients.

## Method

### Data source

This was a retrospective cohort study conducted using data from the National Health Insurance Research Database (NHIRD) of Taiwan, which was described in a previous study [18]. The NHIRD contains all diagnosis, treatment, medication, and other claims data for outpatient and inpatient visits from all medical facilities contracted with the National Health Insurance Administration. The diagnostic codes are those of the International Classification of Diseases, 9th Revision, Clinical Modification (ICD-9-CM). All identification data are encrypted before the information is released to researchers. The Institutional Review Board of Linkou Chang Gung Memorial Hospital approved this study.

### Study cohort and design

The study population was patients diagnosed with end-stage kidney disease who received permanent dialysis between January 1, 2004, and December 31, 2012. Permanent dialysis

was verified by ensuring the patients had received approval for a catastrophic illness certificate (ICD-9-CM: 585). Because the reasons of patients with out-of-hospital cardiac arrest sometimes were not clear when coming to emergency department and the medical treatments on the ambulance were not reported in NHIRD [19]. Therefore, the study cohort was limited to ESKD patients who experienced a first in-hospital CPCR after dialysis initiation.

Of the 93,887 ESKD patients initially extracted from the NHIRD, 99 were excluded because of missing information, 415 were excluded because they were aged under 20 years at the time of dialysis initiation, 8,850 were excluded because their follow-up duration was less than 3 months after dialysis initiation, and 77,407 were excluded because they did not experience in-hospital CPCR after dialysis initiation. In our country, only patients with Do Not Resuscitate were not performed CPCR. Furthermore, among the ESKD patients with the first in-hospital CPCR, 728 were excluded because they had been diagnosed with a malignancy before CPCR and 31 were excluded because they had received a kidney transplant before CPCR. Some ESKD patients with malignancy had signed Do Not Resuscitate and also poor clinical outcomes, which were based on cancer types and stages. Besides, patients with kidney transplant had the experience to take immunosuppressive drugs, which might affected cardiovascular risks [20]. Finally, the remaining 6,357 ESKD patients with a first in-hospital CPCR after dialysis initiation were eligible for analysis (**Fig 1**).

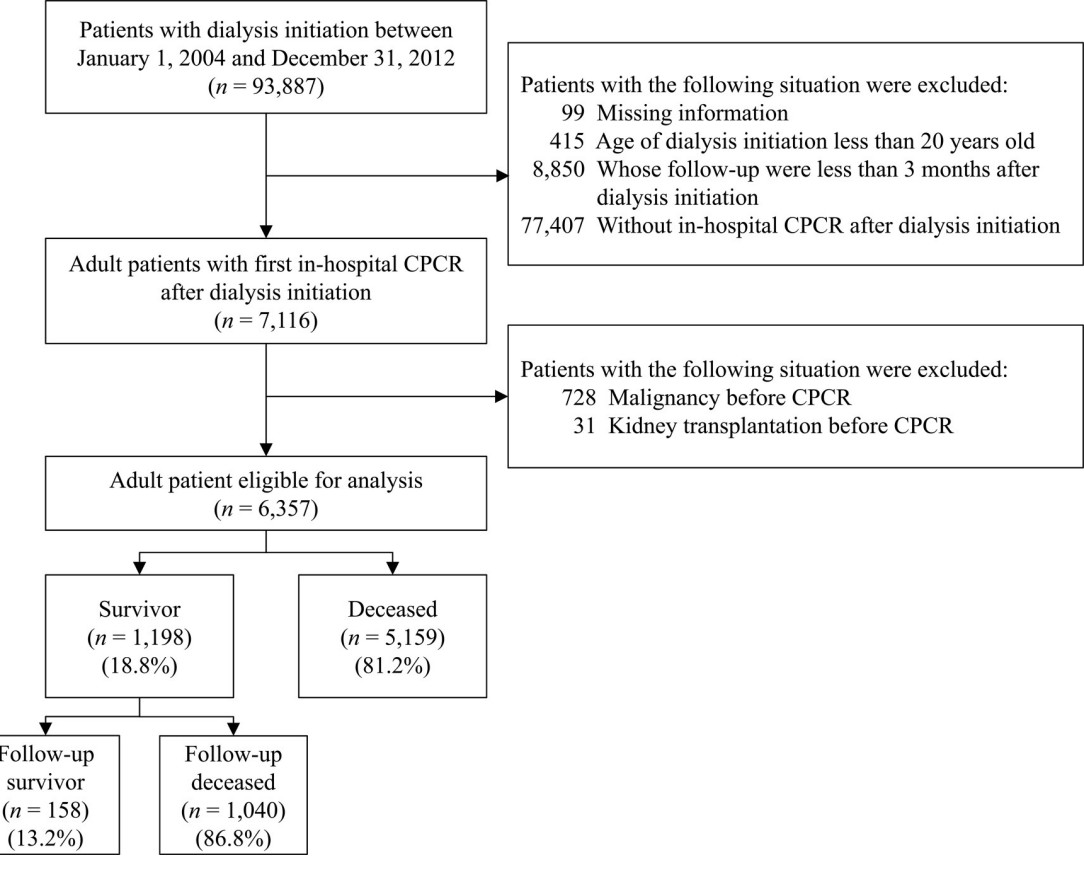

**Fig 1. Patient selection criteria.**

## Study outcome

The primary outcome was the survival rate during the first in-hospital CPCR. The secondary outcome was the median post-discharge survival. We identified all hospital admissions and all in-hospital CPCR events that occurred beyond 3 months after initial dialysis based on the National Health Insurance (NHI) reimbursement codes. We did not include CPCR events that occurred in emergency departments. In-hospital death was defined as withdrawal from the NHI system within 7 days of the discharge date of the index in-hospital CPCR. All-cause mortality was also defined as withdrawal from the NHI program. Patients who survived the index hospitalization with CPCR were followed from the date of discharge until December 31, 2013, or the date of death, whichever came first.

## Patient characteristics

We extracted the characteristics of the study cohort at the time of first CPCR admission. The data included sex, age, date of initial dialysis, and comorbidities, namely hypertension (ICD-9-CM: 401–405), diabetes mellitus (ICD-9-CM: 250), dyslipidemia (ICD-9-CM: 272), atrial fibrillation (ICD-9-CM: 42731), coronary artery disease (ICD-9-CM: 410–414), peripheral arterial disease (ICD-9-CM: 4400, 4402, 4403, 4408, 4409, 443, 4440, 44422, 4448, 4478, and 4479), and prior ICD or cardiac resynchronization therapy defibrillator (CRT-D) implantation. Hospitalization history and medication before the first in-hospital CPCR were also included. The comorbidities were defined as at least 2 outpatient visits or one inpatient diagnosis in the year before the index in-hospital CPCR. Hospitalization history could be tracked to 1997. Prior ICD or CRT-D implantation was determined using NHI reimbursement codes. Medication usage was extracted for the 3 months before the index in-hospital CPCR.

## Statistical analysis

The baseline characteristics of the patients who died during the in-hospital CPCR (the deceased group) and those who survived the admission (the survivor group) were compared using the *t* test for continuous variables and the chi-square test for categorical variables. The incidence of CPCR was estimated using the total number of in-hospital CPCR events per 1,000 admission days of all ESKD patients. Multiple CPCR events in the same patients in the different days contributed to incidence calculation. The trend of CPCR incidence across the study years was analyzed using a univariate Poisson regression in which the study year was the continuous explanatory variable and the logarithm of total admissions days was the offset variable. The trend of surviving the CPCR admission across the study years was analyzed using a univariate logistic regression. The trend of median post-discharge survival across the study years was assessed using the Jonckheere–Terpstra test. Finally, factors associated with risk of in-hospital death for CPCR were investigated using a multivariable logistic regression analysis in which all the baseline characteristics were treated as explanatory variables [21].

A 2-sided *P* value <0.05 was considered to be statistically significant and no adjustment of multiple testing (multiplicity) was made. The Jonckheere–Terpstra test was a one-sided statistical test. All statistical analyses were performed using SAS version 9.4 (SAS Institute, Cary, NC)

# Results

## Baseline characteristics

Table 1 presents the demographic data, including comorbidities, hospitalization history, and medications, for the entire cohort and stratified by survivor or deceased during the index

**Table 1. Baseline characteristics for entire cohort and stratified by survivor or deceased during the index admission.**

| Variable | Entire cohort ($n = 6,357$) | Survivor ($n = 1,198$) | Deceased ($n = 5,159$) | *P* value |
|---|---|---|---|---|
| Demographic information | | | | |
| Age (years) | 69.5 ± 12.1 | 67.0 ± 12.2 | 70.1 ± 12.0 | <0.001 |
| Age ≥ 65 years, n (%) | 4,241 (66.7) | 706 (58.9) | 3,535 (68.5) | <0.001 |
| Male sex, n (%) | 3,244 (51.0) | 595 (49.7) | 2,649 (51.3) | 0.294 |
| Comorbidities in the previous year, n (%) | | | | |
| Hypertension | 5,474 (86.1) | 1,039 (86.7) | 4,435 (86.0) | 0.492 |
| Diabetes mellitus | 4,565 (71.8) | 905 (75.5) | 3,660 (70.9) | 0.001 |
| Dyslipidemia | 1,140 (17.9) | 245 (20.5) | 895 (17.3) | 0.012 |
| Atrial fibrillation | 536 (8.4) | 82 (6.8) | 454 (8.8) | 0.028 |
| Peripheral arterial disease | 994 (15.6) | 197 (16.4) | 797 (15.4) | 0.393 |
| Coronary artery disease | 3,257 (51.2) | 634 (52.9) | 2,623 (50.8) | 0.195 |
| ICD or CRT-D implantation | 22 (0.3) | 14 (1.2) | 8 (0.2) | <0.001 |
| Charlson Comorbidity Index score | 5.6 ± 2.0 | 5.6 ± 2.0 | 5.6 ± 2.0 | 0.294 |
| Hospitalization history, n (%) | | | | |
| Heart failure | 3,093 (48.7) | 569 (47.5) | 2,524 (48.9) | 0.373 |
| Stroke | 2,188 (34.4) | 394 (32.9) | 1,794 (34.8) | 0.216 |
| Myocardial infarction | 1,138 (17.9) | 203 (16.9) | 935 (18.1) | 0.338 |
| Infection-related hospitalization | 5,542 (87.2) | 1,033 (86.2) | 4,509 (87.4) | 0.274 |
| Medication, n (%) | | | | |
| Aspirin/clopidogrel | 3,503 (55.1) | 693 (57.8) | 2,810 (54.5) | 0.034 |
| ACEI/ARB | 3,054 (48.0) | 604 (50.4) | 2,450 (47.5) | 0.068 |
| β-blocker | 3,023 (47.6) | 607 (50.7) | 2,416 (46.8) | 0.017 |
| Loop diuretics | 2,276 (35.8) | 422 (35.2) | 1,854 (35.9) | 0.643 |
| K-sparing diuretics | 221 (3.5) | 38 (3.2) | 183 (3.5) | 0.523 |
| Statin | 1,598 (25.1) | 330 (27.5) | 1,268 (24.6) | 0.033 |
| Sodium bicarbonate | 180 (2.8) | 35 (2.9) | 145 (2.8) | 0.835 |
| Calcium supplementation | 1,399 (22.0) | 273 (22.8) | 1,126 (21.8) | 0.469 |
| Follow-up duration (years) | 0.3 ± 1.1 | 1.6 ± 2.1 | NA | NA |

ACEI, angiotensin converting enzyme inhibitor; ARB, angiotensin receptor blocker; CRT-D, cardiac resynchronization therapy defibrillator; ICD, intracardiac defibrillator; NA, not applicable.

Data are presented as frequency (percentage) or mean ± standard deviation.

admission. The survival rate from the first in-hospital CPCR to discharge was 18.8% (1,198 of 6,357 patients). The patients in the survivor group were on average younger than those in the deceased group. Compared with the deceased group, the survivor group had a significantly higher prevalence of diabetes mellitus, dyslipidemia, and ICD or CRT-D implantation and a lower prevalence of atrial fibrillation. Regarding medication use in the previous 3 months, the patients in the survivor group were more likely to be prescribed aspirin/clopidogrel, beta blockers, and statins than were those in the deceased group.

## Incidence of CPCR and median survival of post-discharge survival

From 2004 through 2012, the incidence of CPCR decreased from 3.97 to 3.67 events per 1,000 admission days, respectively (*P* for trend <0.001) (**Fig 2**; detailed data in **S1 Table**). The survival rate from the first in-hospital CPCR did not change significantly across the 9 years (*P* for trend = 0.244), whereas the median survival of post-discharge survival increased significantly

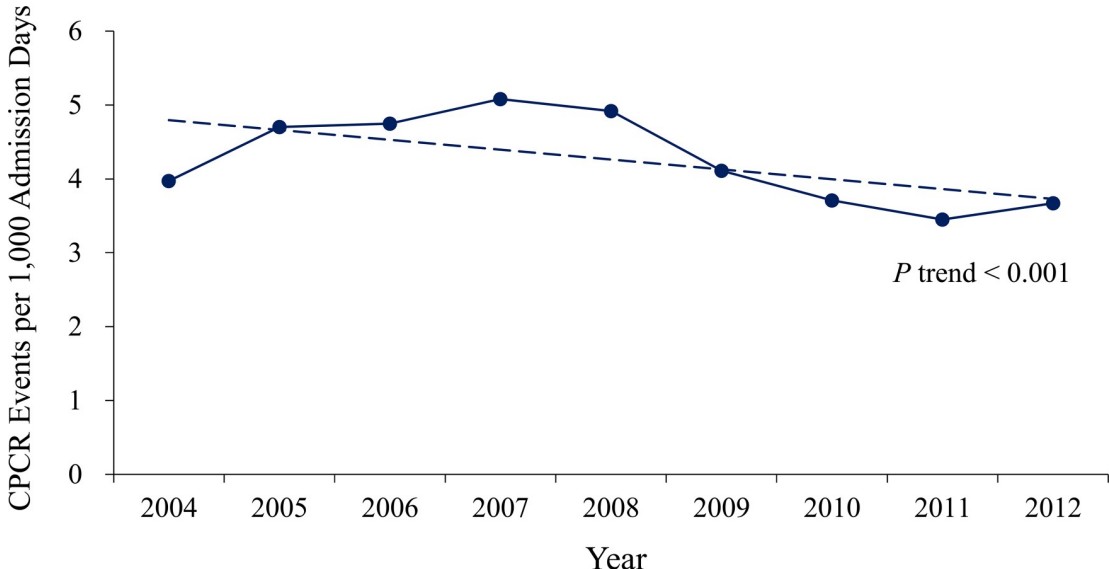

**Fig 2. Incidence of Cardiopulmonary Cerebral Resuscitation (CPCR) across the study years.**

from 3.0 months in 2004 to 6.8 months in 2011 (*P* for trend <0.001) (**Fig 3**; detailed data in **S2 Table**).

### Associated factors of in-hospital death of CPCR

**Table 2** shows the findings of multivariable logistic regression analysis, including all baseline characteristics, for predicting in-hospital CPCR death. The result revealed that an older patient age (odds ratio 1.21, 95% confidence interval [CI] 1.15–1.28) was a risk factor for in-hospital death, while the presence of ICD or CRT-D implantation (odds ratio 0.15, 95% CI 0.06–0.37) was a protective factor for in-hospital death.

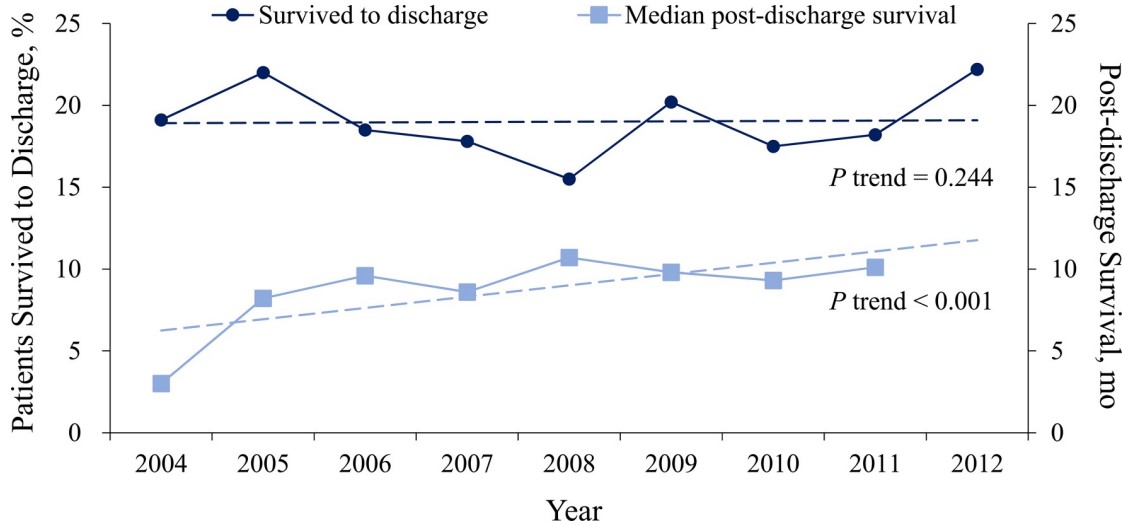

**Fig 3. CPCR admission survival and median post-discharge survival across the study years.** The median survival month after discharge was not calculated in 2012 due to insufficient potential follow-up.

**Table 2. Factors associated with risks of in-hospital death of CPCR.**

| Variable | Odds ratio | 95% CI | *P* value |
|---|---|---|---|
| Age (per 10 years) | 1.21 | 1.15–1.28 | <0.001 |
| Male sex | 1.13 | 0.99–1.28 | 0.074 |
| Comorbidities | | | |
| Hypertension | 0.997 | 0.82–1.21 | 0.977 |
| Diabetes mellitus | 0.86 | 0.74–1.01 | 0.060 |
| Dyslipidemia | 0.91 | 0.76–1.08 | 0.280 |
| Atrial fibrillation | 1.19 | 0.93–1.53 | 0.170 |
| Peripheral arterial disease | 0.99 | 0.83–1.18 | 0.905 |
| Coronary artery disease | 0.93 | 0.80–1.07 | 0.318 |
| ICD or CRT-D implantation | 0.15 | 0.06–0.37 | <0.001 |
| Hospitalization history | | | |
| Heart failure | 1.05 | 0.92–1.20 | 0.476 |
| Stroke | 1.06 | 0.92–1.21 | 0.444 |
| Myocardial infarction | 1.16 | 0.97–1.39 | 0.108 |
| Infection-related hospitalization | 1.07 | 0.89–1.30 | 0.460 |
| Medication | | | |
| Aspirin/clopidogrel | 0.89 | 0.77–1.03 | 0.114 |
| ACEI/ARB | 0.97 | 0.84–1.11 | 0.654 |
| β-blocker | 0.96 | 0.84–1.10 | 0.571 |
| Loop diuretics | 1.06 | 0.92–1.23 | 0.403 |
| K-sparing diuretics | 1.16 | 0.81–1.67 | 0.423 |
| Statin | 1.02 | 0.86–1.20 | 0.857 |
| Sodium bicarbonate | 0.93 | 0.63–1.36 | 0.690 |
| Calcium supplementation | 0.98 | 0.83–1.14 | 0.748 |

ACEI, angiotensin converting enzyme inhibitor; ARB, angiotensin receptor blocker; CI, confidence interval; CPCR, cardiopulmonary cerebral resuscitation; CRT-D, cardiac resynchronization therapy defibrillator; ICD, intracardiac defibrillator.

## Discussion

This study yielded four major findings. First, from 2004 through 2013, the incidence of in-hospital CPCR for ESKD patients decreased significantly, falling from 3.97 to 3.67 events per 1,000 admission days. Second, although the in-hospital survival rate did not exhibit improvement, the median months of post-discharge survival increased significantly to 6.8 months. Third, multivariable analysis revealed that only age and ICD or CRT-D implantation were strong predictors for successful CPCR and that the traditional risk factors for cardiac death were statistical nonsignificant. Finally, neither beta blockers nor ACEI/ARB was associated with a survival benefit.

### Previous studies

The outcomes of ESKD patients after in-hospital CPCR have rarely been reported in previous studies. Of the relevant research present in the literature, much of the data are older and/or based on small-population studies, rendering them inadequate for properly reflecting the current CPCR algorithm in wide use. The findings of previous related studies are summarized in S3 Table. As the data show, the prognosis after CPCR in ESKD patients is poor, with a discharge alive rate ranging from 0% to 26.1% in the studies (excluding that of Lafrance et al.

[22]). Three studies [7, 23, 24] have confirmed the prognosis of cardiac arrest or acute coronary syndrome as being worse in ESKD patients than in the general population. Wong et al. [21] found that the incidence of in-hospital CPCR increased from 2000 through 2010 but that the survival rate improved in the later years, similar to the findings of Saeed et al. [7]. Herzog et al. found that the incidence of cardiac arrest increased along with dialysis vintage (cardiac arrest rates of 93 and 164 events/1,000 patient years at years 1 and 4, respectively), with a 1-year survival rate of 15%. Diabetes mellitus was found to be a risk factor and poor outcome predictor for sudden cardiac arrest in their study [6]. As mentioned, the clinic outcomes of ESKD patients improved in previous study but remained poorer than those of the general population. Therefore, determining possible protective factors for dialysis patients after CPCR should be a crucial goal of future research.

## Current study

Compared with that of the U.S.-based study by Wong et al., the incidence of in-hospital CPCR for ESKD patients was higher in the current Taiwan-based study, even after the incidence decreased significantly to 3.67 CPR events per 1,000 admission days in 2012. From 2004 to 2008, there was a small increase in the incidence of in-hospital CPCR in our cohort study; however, the overall trend remained downward from 2004 to 2013. Furthermore, the in-hospital survival rate of ESKD patients after CPCR improved slightly but was statistical nonsignificant, in contrast to the studies of Saeed et al. [23] and Wong et al. [21]. There are 2 possible reasons for a significant decrease in in-hospital CPR. The first is the improvement of overall medical care quality in Taiwan throughout the NHI era. The second is the passage of the Natural Death Act in 2000, which was aimed at avoiding unnecessary CPR in terminal patients. The increase we observed in median months of post-discharge survival suggest that survival and discharge to home status were better in 2012 than in previous years. Available data from the general population also indicate that survival and neurologic status after CPR are improving, probably as a result of better resuscitation care and CPR techniques [23, 25].

In line with previous research, the present study used multivariable regression analysis and confirmed Pun's observation [15], indicating that traditional cardiovascular factors including diabetes mellitus, hypertension, dyslipidemia, and underlying coronary artery disease might not be correlated with survival after cardiac arrest. This result may be explained by considering that the leading cause of death in dialysis is sudden cardiac death and arrhythmia and that the risk factor/pathogenesis of sudden cardiac death in ESKD patient differ from those of cardiovascular death in the general population. These risk factors include cardiac pathophysiology change (e.g., myocardial interstitial fibrosis, microvascular calcification, QT prolongation, left ventricular hypertrophy, and ventricular dysfunction), rapid electrolyte shift, chronic inflammation, uremia, mineral and bone disorder in chronic kidney disease, hypervolemia, and the dialysis procedure itself [8–13, 26, 27].

To our knowledge, this is the first nationwide, population-based study to examine the role of ICD implantation in ESKD patient. Both ICD and CRT-D implantation seemed to be a possible cardioprotective factor in ESKD patients after CPCR in our study. Wan et al. reported that ESKD patients with wearable cardioverter defibrillators had better survival after the event of sudden cardiac arrest [28]. In addition, Herzog at el. found that only 7.6% of ESKD patients who survived a cardiac arrest had received an ICD implantation. Their study also revealed that ICD implantation had a secondary preventive role, namely that it was associated with greater survival in cardiac arrest survivors [16]. A meta-analysis by Chen et al. [29] found that ICD implantation improved overall survival in ESKD patients with heart failure. The abovementioned findings may suggest some beneficial effect of ICD implantation in ESRD patients.

However, data from a prospective, randomized study are lacking and no international consensus exists regarding ICD use in ESKD patients. Two prospective, randomized trials, Implantable Cardioverter Defibrillators in Dialysis Patients (ISRCTN20479861) [30] and Wearable Cardioverter Defibrillator in Hemodialysis Patients (NCT02481206) are currently evaluating the role of ICD in sudden cardiac death prevention in ESKD patients, but results were still impending as of March 2019.

The present study indicated that aspirin/clopidogrel, beta blockers, and statins might be associated with survival after CPCR, but these associations were not statistical significant after multivariable regression analysis. A previous study suggested that the pathogenesis of cardiovascular events in ESKD patients differ from those in the general population [5] and, as mentioned above, the pathogenesis of cardiovascular death in dialysis also differs. Nevertheless, many previous studies have shown the beneficial effects of beta blockers in lowing sudden cardiac mortality and reducing the incidence of sudden cardiac death specifically in ESKD patients [15, 31–33]. The underlying reason or reasons for this remain unclear. One possible explanation is that to avoid intradialytic hypotension, the pre-dialytic dosage of cardiovascular drugs is sometimes reduced by the patient or physician; therefore, the actual dose may be lower than that prescribed.

## Clinical implications

Sudden cardiac death and ventricular arrhythmias are common outcomes for ESKD patients who receive CPCR [34, 35]. In the general population, ICD implantation could reduce mortality in patients with sudden cardiac arrest [36]. The role of ICD implantation in the primary prevention of sudden cardiac death in ESKD patients is obscure and the data from 2 large prospective randomized studies that may shed further light on the underlying mechanism are still not yet available. The results of our cohort study lend support to the notion that ICD implantation has a survival benefit in ESKD patients after CPCR. These results imply that ICD implantation might improve clinical outcomes for ESKD patients who are at high risk of sudden cardiac death or otherwise clinically indicated.

## Study limitations

This study was based on data from a large administrative database and thus its design has several limitations. First, no personal data such as family history and lifestyle or laboratory data for parameters including creatinine level, BP records, or lipid data were available. Therefore, the etiology of in-hospital CPCR, such as the details of the cardiac arrest events including the initial cardiac arrest rhythm, witness status, bystander status and treatments used during and after resuscitation, could not be included. Not like patients with OHCA, witness status and bystander status were less important for patients with in-hospital cardiac arrest [22, 37, 38]. Second, this cohort study only included ESKD patients with in-hospital CPCR and didn't include ESKD patient with CPCR in emergency departments. Therefore, the extrapolation validity was limited to ESKD patients with in-hospital CPCR and selection bias may exist. Third, we did not have data regarding the patients' status after in-hospital CPCR, such as their levels of disability or nursing home use. Fourth, the clinical indications for patients using beta blockers were unavailable, as was their actual and prescribe doses. Finally, we were unable to take repeat CPCR in the same day into account.

## Conclusion

The incidence of in-hospital CPCR and the survival duration post-discharge among ESKD patients improved despite there being no significant difference in the survival rate of ESKD

patients after CPCP. Either ICD or CRT-D implantation may be advisable for selected dialysis patients with a high risk of sudden cardiac death or with clinical indication.

## Supporting information

**S1 Table. Admitted CPCR events per 1,000 admission days of admitted ESKD patients.**
(DOCX)

**S2 Table. Median survival months.**
(DOCX)

**S3 Table. Summary of studies about ESKD patient post CPCR outcome.**
(DOCX)

## Author Contributions

**Data curation:** Jih-Kai Yeh.

**Formal analysis:** George Kuo.

**Investigation:** Cheng-Chia Lee.

**Methodology:** I-Chang Hsieh.

**Resources:** Ming-Jer Hsieh.

**Software:** Ya-Chung Tian.

**Supervision:** Chih-Hsiang Chang.

**Writing – original draft:** Chia-Hung Yang.

**Writing – review & editing:** Jia-Jin Chen.

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
