## [Decision Letter · Decision Letter 0]

28 Jan 2020

PONE-D-19-32280

The Incidence and Survival after In-hospital Cardiopulmonary Cerebral Resuscitation in Dialytic Patients: A Nationwide Population-based Study

PLOS ONE

Dear Dr. Chang,

Thank you for submitting your manuscript to PLOS ONE. After careful consideration, we feel that it has merit but does not fully meet PLOS ONE’s publication criteria as it currently stands. Therefore, we invite you to submit a revised version of the manuscript that addresses the points raised during the review process.

We would appreciate receiving your revised manuscript by Mar 13 2020 11:59PM. To enhance the reproducibility of your results, we recommend that if applicable you deposit your laboratory protocols in protocols.io, where a protocol can be assigned its own identifier (DOI) such that it can be cited independently in the future. For instructions see: http://journals.plos.org/plosone/s/submission-guidelines#loc-laboratory-protocols

We look forward to receiving your revised manuscript.

Kind regards,

Micah Chan

Academic Editor

PLOS ONE

Additional Editor Comments (if provided):

Well written manuscript. Please respond to the reviewers comments and re-submit

Journal Requirements:

2. In ethics statement in the manuscript and in the online submission form, please provide additional information about the patient records used in your retrospective study. Specifically, please ensure that you have discussed whether all data were fully anonymized before you accessed them and/or whether the IRB or ethics committee waived the requirement for informed consent. If patients provided informed written consent to have data from their medical records used in research, please include this information.

3. Thank you for stating the following in the Funding Section of your manuscript: "This study was supported by grants from Chang Gung Memorial Hospital, Taiwan (CMRPG3F1651, CORPG3G0761). Dr. C-H Chang was supported by the Ministry of Science and Technology (106-2314-B-182A-118-MY3)"

4. Your ethics statement must appear in the Methods section of your manuscript. If your ethics statement is written in any section besides the Methods, please move it to the Methods section and delete it from any other section. Please also ensure that your ethics statement is included in your manuscript, as the ethics section of your online submission will not be published alongside your manuscript.

Reviewers' comments:

Reviewer's Responses to Questions

**Comments to the Author**

1. Is the manuscript technically sound, and do the data support the conclusions?

Reviewer #1: Yes

Reviewer #2: Yes

2. Has the statistical analysis been performed appropriately and rigorously? 

Reviewer #1: Yes

Reviewer #2: Yes

3. Have the authors made all data underlying the findings in their manuscript fully available?

Reviewer #1: No

Reviewer #2: Yes

4. Is the manuscript presented in an intelligible fashion and written in standard English?

Reviewer #1: Yes

Reviewer #2: Yes

5. Review Comments to the Author

Reviewer #1: This is a very interesting retrospective cohort study from Taiwan of incidence and survival after in-hospital cardiopulmonary cerebral resuscitation in ESKD patients. It identified key risk factors and key protective factors in this critical demographic. This manuscript provides an interesting look into such a well-known complication in such a high-risk population. However, the article needs to be specific in stating that they looked at ESKD population and not all dialytic patients. There are some points that need to be clearly defined. There are some grammatical errors that need to be addressed as well.

Please see my attached file for full comments.

Reviewer #2: Outside of some minor grammatical errors and syntax, this is a well written manuscript with conclusions that are of interest to the readership of the Journal. I congratulate the authors on their findings.

6. PLOS authors have the option to publish the peer review history of their article (what does this mean?). If published, this will include your full peer review and any attached files.

Reviewer #1: No

Reviewer #2: No

---

## [Author Response · Author response to Decision Letter 0]

7 Apr 2020

Dear Professor Joerg Heber and reviewers, 

 We are pleased that you offered us an invitation to revise our work entitled “The Incidence and Survival after In-hospital Cardiopulmonary Cerebral Resuscitation in End-Stage Kidney Disease Patients: A Nationwide Population-based Study” for PLOS ONE. The comments and insights were a tremendous help to us during this revision. In the revised manuscript, in accordance with the valuable suggestions of the reviewers, we have made some modifications. The major concerns of the reviewers have been fully addressed, and the entire manuscript has been carefully revised according to the instructions as directed by PLOS ONE. We had also tried our best to address the issues of reviewer #1. All the revised words and sentences are marked as red color. We hope that the revised manuscript will retain your attention and you will judge the revised manuscript to be suitable for publication in PLOS ONE. 

Thanks again for reviewing our manuscript.

Yours sincerely,

First author: Chia-Hung Yang, 

Corresponding author: Chih-Hsiang Chang

 

Reviewer #1: This is a very interesting retrospective cohort study from Taiwan of incidence and survival after in-hospital cardiopulmonary cerebral resuscitation in ESKD patients. It identified key risk factors and key protective factors in this critical demographic. This manuscript provides an interesting look into such a well-known complication in such a high-risk population. However, the article needs to be specific in stating that they looked at ESKD population and not all dialytic patients. There are some points that need to be clearly defined. There are some grammatical errors that need to be addressed as well.

Answer: We thank for your suggestion. We have revised several sentences to make some points to be clearly defined

Reviewer #2: Outside of some minor grammatical errors and syntax, this is a well written manuscript with conclusions that are of interest to the readership of the Journal. I congratulate the authors on their findings.

Answer: Thanks for your great recommendation.

---

## [Decision Letter · Decision Letter 1]

2 Jul 2020

PONE-D-19-32280R1

The Incidence and Survival after In-hospital Cardiopulmonary Cerebral Resuscitation in End-Stage Kidney Disease Patients: A Nationwide Population-based Study

PLOS ONE

Dear Dr. Chang,

Thank you for submitting your manuscript to PLOS ONE. After careful consideration, we feel that it has merit but does not fully meet PLOS ONE’s publication criteria as it currently stands. Therefore, we invite you to submit a revised version of the manuscript that addresses the points raised during the review process.

Specifically, the statistics section should be properly expanded and selection criteria should be discussed. Potential selection biases should be properly acknowledged and addressed in the discussion and in a proper limitation section.

We look forward to receiving your revised manuscript.

Kind regards,

Antonio Cannatà

Academic Editor

PLOS ONE

Additional Editor Comments (if provided):

The authors should properly provide scientific evidence regarding the selection process and careflly describe inclusion and exclusion criteria.

Proper statistics should be performed and a flow-chart with details regarding screened and enrolled patients should be provided.

Reviewers' comments:

Reviewer's Responses to Questions

**Comments to the Author**

1. If the authors have adequately addressed your comments raised in a previous round of review and you feel that this manuscript is now acceptable for publication, you may indicate that here to bypass the “Comments to the Author” section, enter your conflict of interest statement in the “Confidential to Editor” section, and submit your "Accept" recommendation.

Reviewer #3: (No Response)

Reviewer #4: All comments have been addressed

2. Is the manuscript technically sound, and do the data support the conclusions?

Reviewer #3: No

Reviewer #4: Yes

3. Has the statistical analysis been performed appropriately and rigorously? 

Reviewer #3: No

Reviewer #4: Yes

4. Have the authors made all data underlying the findings in their manuscript fully available?

Reviewer #3: Yes

Reviewer #4: Yes

5. Is the manuscript presented in an intelligible fashion and written in standard English?

Reviewer #3: Yes

Reviewer #4: Yes

6. Review Comments to the Author

Reviewer #3: - It is not clear why the authors excluded from the study ESKD patients who did not undergo CPCR. The analysis of the characteristics of patients who did and did not undergo CPCR after a cardiac arrest event would have been of interest and would have helped in determining the external validity of the findings (e.g., do the findings apply to all hospitalized ESKD patients who experience a cardiac arrest, or only to those who undergo CPCR?)

- “We did not include CPCR events that occurred in emergency departments” the reasons for this are not clear but very likely to bias the results of the study

- “All patients were followed from their index admission until December 31, 2013” this gives the idea that patients who were alive at discharge were still followed over time, however the outcome of interest is in-hospital mortality. The authors should clarify this point, December 31 2013 seems to be the administrative censoring of the study

- The authors exclude patients with a diagnosis of malignancy and those with a previous transplant although these criteria are not specified among the inclusion/exclusion criteria. Did the authors generate a statistical analytical plan drafted before data analysis?

- The study lacks a comparison group, be it ESKD patients not undergoing CPCR or non-ESKD patients undergoing CPCR. They perform inferential statistical analyses on trends over calendar year, however this is not particularly interesting nor relevant

- Some of the results reported in Table 2 are puzzling: diabetes seems to be associated with better outcomes, with an OR of 0.86 with a p-value of 0.060. This is indirect suggestion of a potential selection bias in the study

Reviewer #4: Thank you very much for the opportunity to review a manuscript titled “The Incidence and Survival after In-hospital Cardiopulmonary Cerebral Resuscitation in End-Stage Kidney Disease Patients: A Nationwide Population based Study”, which was first reviewed by other reviewers. This is the second review for this manuscript and the first review by me. This manuscript reported survival outcomes and their associated factors for patients with end stage kidney disease who received the first in-hospital cardiopulmonary cerebral resuscitation after the initiation of dialysis. The methods used appear to be valid. The finding gained also appear appropriate. Overall, this manuscript is well written. However, I have one concern. The predictors used in this study does not include important factors for outcomes of patients after cardiac arrest, which include initial arrest rhythm, witness status, and bystander status. As this study included only in-hospital cardiac arrest patients, these factors may not be important compared to out-of-hospital cardiac arrest cases. I understand that the authors briefly described this limitation in the Limitations, and I am not sure if these factors are less important for in-hospital cardiac arrest patients than for OHCA patients, but it would be better if the authors could address this issue by citing other papers. I found some typos as shown below. Therefore, it would be better if this manuscript could be proofread thoroughly again.

Line 6 under a paragraph with a heading ‘Study outcome’: What is ‘NIH’? I presume this is ‘NHI’. If not, please spell this out. The heading should be ‘Study outcomes’ as there are multiple outcomes.

The 2nd last line of the first paragraph of the Statistical analysis: A space is missing between ‘CPCR’ and ‘were’.

In the paragraph starting with ‘To our knowledge’ in the Discussion: A term ‘metaanalysis’ is less commonly used than a term ‘meta-analysis’.

In the paragraph starting with ‘The present study indicated’ in the Discussion: A clause ‘benefits effect’ sound awkward. I feel it may be ‘beneficial effects’?

7. PLOS authors have the option to publish the peer review history of their article (what does this mean?). If published, this will include your full peer review and any attached files.

Reviewer #3: No

Reviewer #4: No

---

## [Author Response · Author response to Decision Letter 1]

4 Aug 2020

Dear Professor Joerg Heber and reviewers, 

 We are pleased that you offered us an invitation to revise our work entitled “The Incidence and Survival after In-hospital Cardiopulmonary Cerebral Resuscitation in End-Stage Kidney Disease Patients: A Nationwide Population-based Study” for PLOS ONE. The comments and insights were a tremendous help to us during this revision. In the revised manuscript, in accordance with the valuable suggestions of the reviewers, we have made some modifications. The major concerns of the reviewers have been fully addressed, and the entire manuscript has been carefully revised according to the instructions as directed by PLOS ONE. We had also tried our best to address the issues of reviewer #1. All the revised words and sentences are marked as red color. We hope that the revised manuscript will retain your attention and you will judge the revised manuscript to be suitable for publication in PLOS ONE. 

Thanks again for reviewing our manuscript.

Yours sincerely,

First author: Chia-Hung Yang, 

Corresponding author: Chih-Hsiang Chang

 

Additional Editor Comments (if provided):

The authors should properly provide scientific evidence regarding the selection process and carefully describe inclusion and exclusion criteria.

Proper statistics should be performed and a flow-chart with details regarding screened and enrolled patients should be provided.

Ans: We thank for your suggestion. We have modified according to your advice.

Reviewer #3: - It is not clear why the authors excluded from the study ESKD patients who did not undergo CPCR. The analysis of the characteristics of patients who did and did not undergo CPCR after a cardiac arrest event would have been of interest and would have helped in determining the external validity of the findings (e.g., do the findings apply to all hospitalized ESKD patients who experience a cardiac arrest, or only to those who undergo CPCR?)

Ans: Thanks for your comments. In our country, only patients with Do Not Resuscitate were not performed CPCR. One of our primary goal of this study is to investigate the possible risk or protective factors of the risk of in-hospital death for these patients. Hence, we excluded the ESKD patients without undergoing CPCR.

- “We did not include CPCR events that occurred in emergency departments” the reasons for this are not clear but very likely to bias the results of the study

Ans: Thanks for your valuable comments. The study cohort was ESKD patients who experienced a first in-hospital CPCR after dialysis initiation. Because the reasons of patients with out-of-hospital cardiac arrest sometimes were not clear when coming to emergency department and the medical treatments on the ambulance were not reported in NHIRD. Therefore, the extrapolation validity was limited to ESKD patients with in-hospital CPCR. We have described this issue in the “Study cohort and design” subsection and added this to the second point of the limitations.

- “All patients were followed from their index admission until December 31, 2013” this gives the idea that patients who were alive at discharge were still followed over time, however the outcome of interest is in-hospital mortality. The authors should clarify this point, December 31 2013 seems to be the administrative censoring of the study

Ans: We agreed with your assessment. We have revised several sentences. “Patients who survived the index hospitalization with CPCR were followed from the date of discharge until December 31, 2013, or the date of death, whichever came first.” 

- The authors exclude patients with a diagnosis of malignancy and those with a previous transplant although these criteria are not specified among the inclusion/exclusion criteria. Did the authors generate a statistical analytical plan drafted before data analysis?

Ans: Thanks for your comments. We revised our inclusion and exclusion criteria, as shown in the second paragraph of the “Study cohort and design” subsection. Our statistical analytical plan generally referred to the reference 21. 

- The study lacks a comparison group, be it ESKD patients not undergoing CPCR or non-ESKD patients undergoing CPCR. They perform inferential statistical analyses on trends over calendar year, however this is not particularly interesting nor relevant

Ans: Thanks for your comment. In this cohort study, we focused the survival and incidence of ESKD patients with CPCR. We wanted to understand the improvement of medical care of ESKD patients with in-hospital CPCR. Figure 2 and Figure 3 show the improvement of care quality in our country over time. Therefore, the results were very important and useful for clinical doctors when discussing the medical conditions with ESKD patients and their families. Based on this, we prefer to retain this epidemiological information. 

- Some of the results reported in Table 2 are puzzling: diabetes seems to be associated with better outcomes, with an OR of 0.86 with a p-value of 0.060. This is indirect suggestion of a potential selection bias in the study

Ans: We thank for your suggestion. We agree with you and added this to the second point of the limitations. 

Reviewer #4: Thank you very much for the opportunity to review a manuscript titled “The Incidence and Survival after In-hospital Cardiopulmonary Cerebral Resuscitation in End-Stage Kidney Disease Patients: A Nationwide Population based Study”, which was first reviewed by other reviewers. This is the second review for this manuscript and the first review by me. This manuscript reported survival outcomes and their associated factors for patients with end stage kidney disease who received the first in-hospital cardiopulmonary cerebral resuscitation after the initiation of dialysis. The methods used appear to be valid. The finding gained also appear appropriate. Overall, this manuscript is well written. However, I have one concern. The predictors used in this study does not include important factors for outcomes of patients after cardiac arrest, which include initial arrest rhythm, witness status, and bystander status. As this study included only in-hospital cardiac arrest patients, these factors may not be important compared to out-of-hospital cardiac arrest cases. I understand that the authors briefly described this limitation in the Limitations, and I am not sure if these factors are less important for in-hospital cardiac arrest patients than for OHCA patients, but it would be better if the authors could address this issue by citing other papers. I found some typos as shown below. Therefore, it would be better if this manuscript could be proofread thoroughly again.

Ans: Thank you for this valuable comments and references. In this revision, we have added some references and changed the wording based on your suggestion in the “Study Limitations”

Line 6 under a paragraph with a heading ‘Study outcome’: What is ‘NIH’? I presume this is ‘NHI’. If not, please spell this out. The heading should be ‘Study outcomes’ as there are multiple outcomes.

The 2nd last line of the first paragraph of the Statistical analysis: A space is missing between ‘CPCR’ and ‘were’.

In the paragraph starting with ‘To our knowledge’ in the Discussion: A term ‘metaanalysis’ is less commonly used than a term ‘meta-analysis’.

In the paragraph starting with ‘The present study indicated’ in the Discussion: A clause ‘benefits effect’ sound awkward. I feel it may be ‘beneficial effects’?

Answer: Thank you for this valuable comments. We have fixed these wrong words.

---

## [Decision Letter · Decision Letter 2]

10 Aug 2020

The Incidence and Survival after In-hospital Cardiopulmonary Cerebral Resuscitation in End-Stage Kidney Disease Patients: A Nationwide Population-based Study

PONE-D-19-32280R2

Dear Dr. Chang,

We’re pleased to inform you that your manuscript has been judged scientifically suitable for publication and will be formally accepted for publication once it meets all outstanding technical requirements.

Kind regards,

Antonio Cannatà

Academic Editor

PLOS ONE

Additional Editor Comments (optional):

Reviewers' comments:

Reviewer's Responses to Questions

**Comments to the Author**

1. If the authors have adequately addressed your comments raised in a previous round of review and you feel that this manuscript is now acceptable for publication, you may indicate that here to bypass the “Comments to the Author” section, enter your conflict of interest statement in the “Confidential to Editor” section, and submit your "Accept" recommendation.

Reviewer #3: All comments have been addressed

Reviewer #4: All comments have been addressed

2. Is the manuscript technically sound, and do the data support the conclusions?

Reviewer #3: (No Response)

Reviewer #4: Yes

3. Has the statistical analysis been performed appropriately and rigorously? 

Reviewer #3: (No Response)

Reviewer #4: Yes

4. Have the authors made all data underlying the findings in their manuscript fully available?

Reviewer #3: (No Response)

Reviewer #4: Yes

5. Is the manuscript presented in an intelligible fashion and written in standard English?

Reviewer #3: (No Response)

Reviewer #4: Yes

6. Review Comments to the Author

Reviewer #3: (No Response)

Reviewer #4: Thank you very much for the revised version of the manuscript. The authors appropriately responded to my answers and I was satisfied with them. I have no more concern except one below. This is a minor grammatical error to be corrected. Otherwise, this manuscript looks fine to me. I would like to congratulate the authors for completing this study and appreciate the opportunity to review this manuscript.

The third sentence under a heading of ‘Study cohort and design’:

“Because the reasons of patients with out-of-hospital cardiac arrest sometimes were not clear when coming to emergency department and the medical treatments on the ambulance were not reported in NHIRD[19]. Therefore, the study cohort was limited to ESKD patients who Experienced a first in-hospital CPCR after dialysis initiation.” I think this should be “….. in NHIRD[19]; therefore, …..” or just remove ‘Because’. (The latter may be better?)

7. PLOS authors have the option to publish the peer review history of their article (what does this mean?). If published, this will include your full peer review and any attached files.

Reviewer #3: No

Reviewer #4: No

---

## [Editor Report · Acceptance letter]

17 Aug 2020

PONE-D-19-32280R2 

The Incidence and Survival after In-hospital Cardiopulmonary Cerebral Resuscitation in End-Stage Kidney Disease Patients: A Nationwide Population-based Study 

Dear Dr. Chang:

I'm pleased to inform you that your manuscript has been deemed suitable for publication in PLOS ONE. Congratulations! Your manuscript is now with our production department. 

Kind regards, 

on behalf of

Dr. Antonio Cannatà 

Academic Editor

PLOS ONE